



# Earth Virtualization Engines (EVE)

Bjorn Stevens[1], Stefan Adami[2,3], Tariq Ali[4], Hartwig Anzt[5,6], Zafer Aslan[7], Sabine Attinger[8], Jaana Bäck[9], Johanna Baehr[10], Peter Bauer[11,*], Natacha Bernier[12], Bob Bishop[13], Hendryk Bockelmann[14], Sandrine Bony[15], Veronique Bouchet[12], Guy Brasseur[1], David N. Bresch[16,17], Sean Breyer[18], Gilbert Brunet[12], Pier Luigi Buttigieg[19], Junji Cao[20], Christelle Castet[21], Yafang Cheng[22], Ayantika Dey Choudhury[23], Deborah Coen[24], Susanne Crewell[25], Atish Dabholkar[26], Qing Dai[27], Francisco Doblas-Reyes[28], Dale Durran[29], Ayoub El Gaidi[30], Charlie Ewen[31], Eleftheria Exarchou[32], Veronika Eyring[33,34], Florencia Falkinhoff[35], David Farrell[36], Piers M. Forster[37], Ariane Frassoni[38], Claudia Frauen[14], Oliver Fuhrer[17], Shahzad Gani[39], Edwin Gerber[40], Debra Goldfarb[41], Jens Grieger[42], Nicolas Gruber[16], Wilco Hazeleger[43], Rolf Herken[44], Chris Hewitt[12], Torsten Hoefler[16], Huang-Hsiung Hsu[45], Daniela Jacob[46,47], Alexandra Jahn[48], Christian Jakob[49], Thomas Jung[19], Christopher Kadow[14], In-Sik Kang[50], Sarah Kang[51,1], Karthik Kashinath[52], Katharina Kleinen-von Königslöw[10], Daniel Klocke[1], Uta Kloenne[53], Milan Klöwer[54], Chihiro Kodama[55], Stefan Kollet[56], Tobias Kölling[1], Jenni Kontkanen[57], Steve Kopp[18], Michal Koran[58], Markku Kulmala[9], Hanna Lappalainen[9], Fakhria Latifi[59], Bryan Lawrence[60], June Yi Lee[61,62], Quentin Lejeun[53], Christian Lessig[63], Chao Li[1], Thomas Lippert[56], Jürg Luterbacher[12], Pekka Manninen[57], Jochem Marotzke[1], Satoshi Matsouoka[64], Charlotte Merchant[65], Peter Messmer[52], Gero Michel[66], Kristel Michielsen[56], Tomoki Miyakawa[67], Jens Müller[68], Ramsha Munir[59], Sandeep Narayanasetti[23], Ousmane Ndiaye[69], Carlos Nobre[70], Achim Oberg[10], Riko Oki[71], Tuba Özkan-Haller[72], Tim Palmer[73], Stan Posey[52], Andreas Prein[74], Odessa Primus[58], Mike Pritchard[52], Julie Pullen[75], Dian Putrasahan[1], Johannes Quaas[76], Krishnan Raghavan[23], Venkatachalam Ramaswamy[65,77], Markus Rapp[33], Florian Rauser[78], Markus Reichstein[79], Aromar Revi[80], Sonakshi Saluja[81,82], Masaki Satoh[67], Vera Schemann[25], Sebastian Schemm[16], Christina Schnadt Poberaj[16], Thomas Schulthess[16,83], Cath Senior[31], Jagadish Shukla[84], Manmeet Singh[23], Julia Slingo[31,*], Adam Sobel[85], Silvina Solman[86,87], Jenna Spitzer[43], Detlef Stammer[10], Philip Stier[73], Thomas Stocker[88], Sarah Strock[89], Hang Su[20], Petteri Taalas[12], John Taylor[90], Susann Tegtmeier[91], Georg Teutsch[58], Adrian Tompkins[26], Uwe Ulbrich[42], Pier-Luigi Vidale[60,92], Chien-Ming Wu[93], Hao Xu[27], Najibullah Zaki[59], Laure Zanna[40], Tianjun Zhou[20], and Florian Ziemen[14]

[1]Max Planck Institute for Meteorology, Germany
[2]Technical University of Munich, Germany
[3]AI engineering GmbH, Germany
[4]University of Liverpool, United Kingdom
[5]University of Tennessee, United States
[6]Karlsruhe Institute of Technology, Germany
[7]Istanbul Aydın University, Turkey
[8]Helmholtz Centre for Environmental Research - UFZ, Germany
[9]University of Helsinki, Finland
[10]University of Hamburg, Germany
[11]Destination Earth, European Centre for Medium-Range Weather Forecasts, United Kingdom
[12]World Meteorological Organization, Switzerland
[13]International Centre for Earth Simulation Foundation, Switzerland





[14]German Climate Computing Center (DKRZ), Germany
[15]LMD/IPSL, CNRS, Sorbonne University, France
[16]ETH Zurich, Switzerland
[17]Federal Office of Meteorology and Climatology (MeteoSwiss), Switzerland
[18]Esri Inc., United States
[19]Alfred Wegener Institute for Polar and Marine Research, Germany
[20]Institute of Atmospheric Physics - Chinese Academy of Sciences, China
[21]AXA Climate, France
[22]Max Planck Institute for Chemistry, Germany
[23]Indian Institute of Tropical Meteorology, India
[24]Yale University, United States
[25]University of Cologne, Germany
[26]Abdus Salam International Centre for Theoretical Physics, Italy
[27]Tencent, China
[28]Barcelona Supercomputing Center (BSC), Spain
[29]University of Washington, United States
[30]National Centre of Meteorology, United Arab Emirates
[31]Met Office, United Kingdom
[32]MITIGA Solutions, Spain
[33]German Aerospace Center, Germany
[34]University of Bremen, Germany
[35]Max Planck Institute for Dynamics and Self-Organization, Germany
[36]Caribbean Institute for Meteorology & Hydrology, Barbados
[37]University of Leeds, United Kingdom
[38]Brazilian National Institute for Space Research, Brazil
[39]Indian Institute of Technology Delhi, India
[40]New York University, United States
[41]Amazon Web Services, United States
[42]Freie Universität Berlin, Germany
[43]Utrecht University, Netherlands
[44]MINE Inc., United States
[45]Research Center for Environmental Changes - Academia Sinica, Taiwan
[46]Helmholtz-Zentrum Hereon, Germany
[47]Climate Service Center Germany (GERICS), Germany
[48]University of Colorado Boulder, United States
[49]Monash University, Australia
[50]Indian Ocean Center / Second Institute of Oceanography, India
[51]Ulsan National Institute of Science and Technology, South Korea
[52]NVIDIA Corporation, United States
[53]Climate Analytics, Germany
[54]Massachusetts Institute of Technology, United States
[55]Japan Agency For Marine-Earth Science and Technology, Japan
[56]Forschungszentrum Jülich, Germany
[57]CSC - IT Center for Science, Finland
[58]Global Arena Research Institute, Czech Republic
[59]Technical University of Darmstadt, Germany
[60]University of Reading, United Kingdom
[61]Pusan National University, South Korea
[62]IBS Center for Climate Physics, South Korea



⁶³Otto-von-Guericke-Universität Magdeburg, Germany
⁶⁴Riken Center for Computational Science, Japan
⁶⁵Princeton University, United States
⁶⁶AIG Reinsurance, United States
⁶⁷The University of Tokyo, Japan
⁶⁸Eberswalde University for Sustainable Development, Germany
⁶⁹National Agency for Civil Aviation and Meteorology, Senegal
⁷⁰Science Panel for the Amazon, Brazil
⁷¹Earth Observation Research Center, JAXA, Japan
⁷²Oregon State University, United States
⁷³University of Oxford, United Kingdom
⁷⁴National Center for Atmospheric Research (NCAR), United States
⁷⁵Propeller Ventures, United States
⁷⁶Leipzig University, Germany
⁷⁷National Oceanic and Atmospheric Administration (NOAA), United States
⁷⁸Federal Office for Radiation Protection, Germany
⁷⁹Max Planck Institute for Biogeochemistry, Germany
⁸⁰Indian Institute for Human Settlements, India
⁸¹Reiner Lemoine Institute, Germany
⁸²Alexander von Humboldt Foundation, Germany
⁸³Swiss National Supercomputing Centre, Switzerland
⁸⁴George Mason University, United States
⁸⁵Columbia University, United States
⁸⁶University of Buenos Aires, Argentina
⁸⁷National Scientific and Technical Research Council, Argentina
⁸⁸University of Bern, Switzerland
⁸⁹Hewlett Packard Enterprise, United States
⁹⁰Commonwealth Scientific and Industrial Research Organisation, Australia
⁹¹University of Saskatchewan, Canada
⁹²National Centre for Atmospheric Science, United Kingdom
⁹³National Taiwan University, Taiwan
*retired

**Correspondence:** Bjorn Stevens (bjorn.stevens@mpimet.mpg.de)

**Abstract.** To manage Earth in the Anthropocene, new tools, new institutions, and new forms of international cooperation will be required. Earth Virtualization Engines are proposed as international federation of centers of excellence to empower all people to respond to the immense and urgent challenges posed by climate change.

**An international federation of centers of excellence to empower all people to respond to the immense and urgent challenges posed by climate change**

Every day, more and more people are waking up to the consequences of Earth's changing climate. Changes in weather, water, and ecosystems are catching communities unprepared, and scientists by surprise. These changes are highlighting how little we know about basic questions, such as for whom the monsoon rains may falter, or even fail; or more generally, whether warming

is causing shifts in atmosphere and ocean circulations and how these may connect to more frequent and intense heatwaves, wildfires and flooding. Climate change is also exposing a fundamental injustice, whereby those least responsible are impacted the most. Efforts to know more, and plan better, are severely under-resourced, and inequitably distributed, deepening both the sense of anxiety and the injustice. This increases the disruptive potential of climate impacts for all. EVE responds to this new reality.

EVE envisions a world where everyone knows how climate and climate change affect them, and where this knowledge empowers them to act. By generating entirely new and inherently better sources of information, EVE strives to catalyze a change in the broader ecosystem of data and services to deliver a just, equitable and scientifically grounded basis for action.

EVE will be made up of international centers of excellence, each accessing outstanding computational and data handling capabilities, and each embedded within the rich and expanding landscape of climate-related data, experiences, and information. This will enable EVE to fill a data space with climate projections of much greater fidelity, with local granularity, globally. And to link these, through a digital commons, to data describing the physical, biological, chemical, and social dimensions of the Earth system. EVE's digital commons will efficiently expose data to new (e.g., generative AI, augmented reality) methods of analysis and proactive information production. This will enable stakeholders to construct and interact with their own climate scenarios. EVE's technical ambition will strengthen the capabilities of all sorts of communities to command new technologies – built on scientific excellence, transparency, and openness across disciplines – to rise to the specific challenges climate change poses for them.

EVE will add modelling capacity beyond the reach of most countries, let alone existing modelling centers. This capacity is urgently needed to improve model fidelity, to assess impacts, and to integrate observations, globally. EVE's digital commons will provide access to software and infrastructure-as-a-service that would otherwise not be available at the necessary scale, nor aligned with emerging technologies. Through a commitment to international cooperation and capacity development, EVE's centers of excellence will extend, accelerate, and further open the generation of climate information of unprecedented quality and salience to enhance climate services globally.

Each of EVE's centers of excellence will:

– apply and advance the best available science to continuously grow and refresh a data space with small ensembles of km-scale multi-decadal global climate projections, juxtaposed with larger ensembles at coarser granularity;

– establish and maintain equitable access to a space of interoperable data and software, through open and secure protocols aligned to global standards and conventions, as part of an emerging ecosystem of planetary data;

– support and encourage the innovative uses of data to generate information particularly on local climate impacts and interactively expose it to users, especially those that would not otherwise have access, and to develop standards and trust in their global use;

– cooperate with existing operational climate services and practitioners at all levels, as well as research infrastructures and programmes in the natural, information and social sciences, to amplify both their own and EVE's impact;



– include a strong component of well-tailored capacity development, outreach, and exchange, to enrich EVE's with local knowledge, and to bridge divides to train and employ new developers and users of climate information globally.

Each EVE center of excellence will require experts to maintain and advance state-of-the-art computing and data facilities, to improve its models (physically and computationally), to support training and capacity development, and to engage with users to enlarge the public sphere of climate information. To meet these requirements, alongside EVE's computing and data demands, will require a funding rate of about € 300 million per year per center. Three to five centers should be sufficient to fill the data space of future projections, and at the same time extend access to, and engage, communities globally in using it. Global governance is key to the creation and sustainability of EVE. This could be in the form of an international treaty, or through coordination of self-governed centers by the WMO, UNEP or other inter-governmental organizations. Independent of the governance model, EVE will be charged with maintaining consistent and open delivery of value to the widest possible user community and to support constructive innovation through scientific excellence.

By comparison with the climate-change impacts it seeks to predict, EVE is asking for a tiny investment. All the more so that it takes advantage of an enormous opportunity. And that is to confidently open the door to new worlds. Worlds where water managers in Bhutan can, with confidence and trust, interactively explore the interplay between adaptation strategies and different scenarios of global climate change. Where agriculturalists in northern India, or managers of blue-carbon mangrove stocks in Bangladesh, can anticipate the implication of Bhutan's choices; and where climatologists working in Cape Town can investigate how it all couples back to influence weather, water and ecosystems globally. EVE will allow impact communities, who link ecological changes to patterns of weather and water, to leverage ever richer descriptions of present and past climates to scenarios for the future, to develop strategies for building resilience. EVE's ability to bring forth such worlds will come as much from the fidelity of its new models, as from the power of its new technologies, as from its ability to equitably engage those who need its information most.

**Frequently Asked Questions**

**Are scientists really 'surprised' by the changes?**

Yes, but not by every change, and by different changes in different ways. To be clear, climate science has long established why Earth is warming, and has developed models that have explained its broad trends and contours. Scientists understand, in general terms, how some forms of extremes might change with warming. What's lacking is a specific understanding of such changes, as is needed to guide adaptation. Often this is because projected changes in atmospheric and oceanic circulations and patterns of precipitation differ from one model to the next for reasons we don't understand. While this makes it hard to form expectations, and hence be surprised, there are a growing number of cases where models do agree with one another, but observed changes behave differently, i.e., surprises. A prominent example is the persistence of La Niña conditions over the past half-century (the present El Niño notwithstanding), others include the slowdown in warming between 1998-2013, or the rapidity of arctic sea-ice decline in 2012.

**What is meant by the 'expanding landscape of climate-related data and information'?**

In addition to the reanalyses, which are often the starting point for global climate data, a great many organizations and agencies collect and disseminate climate-related data directly– from satellites, networks of ground sensors, gliders, floats, and drones, to integrative ecosystem supersites. EVE will strengthen these services by making it easier to access their data and combine them with reanalyses and to climate projections with local granularity, globally, by increasing access to open and interoperable data and software, and through the equitable distribution and use of this capacity.

**What is meant by the phrase 'local granularity, globally'?**

There is no unequivocal definition of local, but many people would associate it with the environment that they can perceive with their senses, and the space over which they typically move under their own power, if not a somewhat larger area. This defines local to be about 1 km, or perhaps between 0.1–10 km. Infrastructures are constructed, lives are lived, and impacts are felt on this scale of 'granularity'. This has been appreciated for some time, and has motivated work in climate services to 'downscale'
models with much reduced (100 km) granularity to finer scale. Hence the phrase 'local granularity, globally' emphasizes the importance of the km-scale for impacts, globally; for observations whose footprint is often on a local scale, globally; and for how important climate processes (ocean eddies, overflows, orographic effects, atmospheric convection) influence much larger-scales, sometimes referred to as upscaling.

**Aren't there already a great number of digital twinning activities, what makes EVE different?**

There are a handful of related activities (e.g., Destination Earth, NVIDIA's Omniverse, and the DITTO Programme of the UN Ocean Decade). EVE draws impetus from them, but is larger in scope. EVE differs also through its focus on establishing a sustainable and equitable global footing that better links to the landscape of climate-related data and information, adds interactivity at scale, and is constituted to enable global capacity building, cooperation, and co-development. Depending on local circumstances, an EVE center could, however, emerge as an outgrowth of an existing activity.

**Isn't EVE adopting a technocratic approach that risks increasing inequity?**

EVE is a technology project, and is rooted in the experience that technology, increasingly through the use of AI, can improve the quality of information while simultaneously lowering the barriers to its access. EVE is aware, however, that this requires the engagement of the users of the information (or their trusted representatives) in its production. That is why EVE places such an emphasis on capacity development and exchange in its centers of excellence, to enable co-production at the forefront of
technology. Ideally EVE would give every country in the world the capability to train people to develop models, AI algorithms, and tailor climate information to meet their needs. This is a tall task, and because it touches on the more difficult question of how humans and social systems interact with new sources of information, EVE also must engage research and researchers from the social and behavioral sciences.

**What role will AI play in EVE'?**

AI is one of EVE's core technologies. It can make models more performant, and thereby fit for greater purpose. EVE distinguishes between *AI Inside*, to refer to what is done to help the models perform better, and *AI On-Top* which describes methods to increase information extraction from model output and data. EVE will greatly expand the scope of AI in every regard. EVE's digital commons anticipates the use of generative AI to enable interactivity (AI On-Top) with large amounts of data, and this is envisioned to be a game changer.

**Does EVE aim to replace existing modelling activities?**

No. EVE is additive and complements existing activities by targeting important scales that would otherwise be out of reach, i.e., EVE's 'local granularity, globally'. EVE emphasizes better information provision, whereas research also targets knowledge creation, and these are symbiotic. EVE thus needs and benefits from ongoing modelling activities, for instance as coordinated by the World Climate Research Programme, but at the same time, it will also support those external research efforts through its

technology development, its ability to set standards, by increasing their access to frontier computing, and by providing career paths for their trainees.

**Why 3-5 centers of excellence?**

A smaller number of centers risks not meeting the ambitious computing requirements to fill the required data space with the required diversity in modelling approaches. Too few centers also make it more difficult to engage a sufficient breadth of

users, thereby limiting access to data and expertise. More centers could help EVE be more inclusive, sample more climate trajectories, and engage more users. However, given each center's need to access a critical mass of human resources for model development, to innovatively develop and maintain infrastructure, etc., the reality of limited human and financial resources, and EVE's novelty, having fewer centers, each with a higher profile, is advantageous.

**How much power would an EVE center consume, and is this sustainable?**

EVE's use of high-performance computing requires substantial electricity resources. Based on practice at some of the world's leading supercomputing facilities, it is estimated that each EVE center would need to access approximately 50 MW of power. The compute resources need not be sourced from a dedicated site but must facilitate interoperability of software and data, maintain computing co-proximate with the largest sources of data. By focusing on the development of just a few centers, and concentrating the powered delivery to access renewables and favor circular economies, for instance through productive use of

'waste' heat, EVE will be exemplary of responsible power production and usage.

**How was EVE's budget estimated?**

The € 300M per center per year price tag was estimated based on the current budget of international organizations whose profile overlap with parts of EVE's remit, and it anticipates a roughly equal split between funding for staff, running costs (mostly power), and investments (hardware procurements).

**Will EVE take away resources from existing efforts?**

EVE relies on a vibrant climate research and services community, and cannot be funded at their expense. EVE centers can be expected to employ some of the leading climate science, climate impact, climate services, and technologists world wide, but in the end this will represent a very small fraction of these workforces. Without simultaneously strengthening ongoing research activities, EVE would loose access to trained staff, would become less innovative, and would fail to adequately understand
and communicate its outcomes. Without simultaneously strengthening existing climate services EVE would loose its ability to connect its data and information provision to the communities it must serve, let alone scale this globally. In this sense, EVE will only be successful if it strengthens climate science and service more broadly.

**What's next?**

The outcomes from the Berlin Summit for EVE will be presented and discussed at the World Climate Research Programme
Open Science Conference in Kigali Africa, in October 2023. We are engaging with the WMO and the organizers of COP28 to obtain their formal support, based on which we would develop an implementation plan and governance structure.

**What would happen without EVE?**

Without EVE urgently needed information for adaptation and resilience building would be of inferior quality, and much less accessible to those that need it most. Some of EVE's key technologies and methodologies may be developed anyway, but more
slowly and then only by, and for, the few who can afford to do so. Climate information for business, finance, and global policy, would continue to proliferate, but would lack standardization, inclusivity, and a compelling tie to the best available science. Without EVE, research laboratories would continue to explore the frontiers of computing, but with diminished access, little participation from the global South, and a reduced ability to link their findings and technologies to inform climate actions. Climate services would continue to do their best to exploit advances in the science to inform users, but would be handicapped
in their efforts. The operational aspect of EVE, the co-production of regularly updated information that matches the rapid pace of innovation, would be lost. The world won't be empty-handed, but it would be left with less, and less trusted, information, leaving many less resilient. Just as profoundly, a chance to engage many more people in new, and more equitable, economies at the nexus of emerging technologies and sustainable development would have been missed.

**Summit Participants**

Many people contributed to the Berlin Summit for EVE. A discussion paper in preparation for the summit was downloaded more than 4500 times. A video presentation of EVE at the CONSTRAIN external assembly a week before the meeting was recorded and viewed more than 700 times. A forum was set up that collected extensive input from scores of people. Impulse for the working meeting was provided by a welcome from Bettina Stark-Watzinger, the German minister for research and education, Jensen Huang, founder and CEO of NVIDIA, the most valuable semiconductor company in the world, Debra

Roberts, Co-Chair of the IPCC Working Group II, and by many other distinguished speakers. Participants in the working aspect of the Berlin Summit sampled diverse backgrounds. Joining the many participants working in the host nation (Germany) were representatives of every continent (other than Antarctica), and more than twenty individual countries. Participants ranged from students, like Charlotte Merchant of Princeton University, who led a breakout group, to Petteri Taalas the Secretary General of the World Meteorological Organization. Many past and present leaders of the IPCC, heads of world-renowned research

institutes, leading figures from technology, scientists from fields as diverse as sociology, behavioral psychology, informatics, and the varied fields of climate and climate-impact science, hydrology, ecology, and finance, as well as practitioners from climate services and operational services contributed and joined as signatories.

**Why *Earth System Science Data***

Fundamentally EVE is about improving the quality of and accessibility of Earth system data and doing so in ways that are just

and equitable. Although many journals have higher profiles, and are the more usual targets for such perspectives, none better reflect and practice the values of EVE.

**Signatories**

1. Bjorn Stevens, Director, Max Planck Institute for Meteorology, DE (corresponding author)

2. Stefan Adami, Technical University of Munich / AI engineering GmbH, DE

3. Tariq Ali, Professor and Pro-Vice-Chancellor for Global Engagement & Partnerships, University of Liverpool, UK

4. Hartwig Anzt, University of Tennessee / Karlsruhe Institute of Technology, US/DE

5. Zafer Aslan, Istanbul Aydın University, TR

6. Sabine Attinger, Helmholtz Centre for Environmental Research (UFZ), DE

7. Jaana Bäck, Professor, University of Helsinki, FI

8. Johanna Baehr, Professor, University of Hamburg, DE

9. Peter Bauer, (Director Destination Earth for ECMWF, retired), UK

10. Natacha Bernier, Director of Earth System Science Research and Innovation, Dept. Science and Innovation WMO

11. Bob Bishop, International Centre for Earth Simulation Foundation, CH



12. Hendryk Bockelmann, German Climate Computing Center (DKRZ), DE

13. Sandrine Bony, LMD/IPSL, CNRS, Sorbonne University, FR

14. Veronique Bouchet, World Meteorological Organization (WMO)

15. Guy Brasseur, Director Emeritus, Max Planck Institute for Meteorology, DE

16. David N. Bresch, ETH Zurich / MeteoSwiss, CH

17. Sean Breyer, ESRI Inc.

18. Gilbert Brunet, Chair of the Scientific Advisory Panel, World Meteorological Organisation (WMO)

19. Pier Luigi Buttigieg, Alfred Wegener Institute, DE

20. Junji Cao, Director General, Institute of Atmospheric Physics - Chinese Academy of Science, CN

21. Christelle Castet, AXA Climate, FR

22. Yafang Cheng, Max Planck Institute for Chemistry, DE

23. Ayantika Dey Choudhury, Indian Institute of Tropical Meteorology - Pune, IN

24. Deborah Coen, Professor of History, Yale University, US

25. Susanne Crewell, Professor, University of Cologne, DE

26. Atish Dabholkar, Director, Abdus Salam International Centre for Theoretical Physics, IT

27. Qing Dai, Tencent, SSV Carbon Neutrality Lab, CN

28. Francisco J. Doblas-Reyes, Barcelona Supercomputing Center (BSC), ES

29. Dale Durran, Professor, University of Washington, US

30. Ayoub El Gaidi, National Centre of Meteorology, Presidential Court, AE

31. Charlie Ewen, Chief Technology Officer, Met Office, UK

32. Eleftheria Exarchou, MITIGA Solutions, ES

33. Veronika Eyring, Professor of Climate Modelling, DLR Institute of Atmospheric Physics & University of Bremen, DE

34. Florencia Falkinhoff, Max Planck Institute for Dynamics and Self-Organization, DE

35. David Farrell, Principle, Caribbean Institute for Meteorology & Hydrology, BB

36. Piers Forster, Professor, University of Leeds, UK

37. Ariane Frassoni, Brazilian National Institute for Space Research, BR

38. Claudia Frauen, German Climate Computing Center (DKRZ), DE

39. Oliver Fuhrer, Federal Institute of Meteorology and Climatology, MeteoSwiss, CH

40. Shahzad Gani, Indian Institute of Technology Delhi, IN

41. Edwin Gerber, Professor, New York University, US

42. Debra Goldfarb, Amazon Web Services, US



43.  Jens Grieger, Institute of Meteorology - Freie Universität Berlin, DE

44.  Nicolas Gruber, Professor, ETH Zurich, CH

45.  Wilco Hazeleger, Professor and Dean, Utrecht University, NE

46.  Rolf Herken, MINE

47.  Chris Hewitt, World Meteorological Organization (WMO)

48.  Torsten Hoefler, Professor, ETH Zurich, CH

49.  Huang-Hsiung Hsu, Research Center for Environmental Changes - Academia Sinica, TW

50.  Daniela Jacob, Helmholtz-Zentrum Hereon, Director, Climate Service Center Germany (GERICS), DE

51.  Alexandra Jahn, University of Colorado Boulder, US

52.  Christian Jakob, Professor, Monash University, AU

53.  Thomas Jung, Alfred Wegener Institute, Helmholtz Center for Polar and Marine Research, DE

54.  Christopher Kadow, German Climate Computing Center (DKRZ), DE

55.  In-Sik Kang, Indian Ocean Center/ Second Institute of Oceanography, IN

56.  Sarah Kang, Ulsan National Institute of Science and Technology, Max Planck Institute for Meteorology, KR/DE

57.  Karthik Kashinath, NVIDIA, US

58.  Katharina Kleinen-von Königslöw, University of Hamburg / CLICCS, DE

59.  Daniel Klocke, Max Planck Institute for Meteorology, DE

60.  Uta Kloenne, Climate Analytics, DE

61.  Milan Klöwer, Massachusetts Institute of Technology, US

62.  Chihiro Kodama, Japan Agency For Marine-Earth Science and Technology, JP

63.  Stefan Kollet, Forschungszentrum Jülich GmbH, DE

64.  Tobias Kölling, Max Planck Institute for Meteorology, DE

65.  Jenni Kontkanen, CSC - IT Center for Science, FI

66.  Steve Kopp, Esri Inc.

67.  Michal Koran, Global Arena Research Institute

68.  Thilo Körkel, New Ground e.K., DE

69.  Markku Kulmala, Professor, University of Helsinki, FI

70.  Hanna Lappalainen, University of Helsinki, FI

71.  Fakhria Latifi, Technical University of Darmstadt, DE

72.  Bryan Lawrence, Professor, NCAS / University of Reading, UK

73.  June-Yi Lee, Professor, Pusan National University / IBS Center for Climate Physics, KR



74. Quentin Lejeune, Climate Analytics, DE

75. Christian Lessig, Otto-von-Guericke-Universität Magdeburg, DE

76. Chao Li, Max Planck Institute for Meteorology, DE

77. Thomas Lippert, Head of Jülich Supercomputing Centre, DE

78. Jürg Luterbacher, Director Science and Innovation / Chief Scientist, World Meteorological Organization (WMO)

79. Pekka Manninen, CSC - IT Center for Science, FI

80. Jochem Marotzke, Director, Max Planck Institute for Meteorology, DE

81. Satoshi Matsuoka, Head, Riken Center for Computational Science, JP

82. Charlotte Merchant, Student, Princeton University, US

83. Peter Messmer, NVIDIA, CH

84. Gero Michel, AIG Reinsurance, BS

85. Kristel Michielsen, Forschungszentrum Jülich GmbH, DE

86. Tomoki Miyakawa, Atmosphere and Ocean Research Institute - The University of Tokyo, JP

87. Samuel Morin, Météo-France - CNRS/CNRM, FR

88. Jens Müller, Eberswalde University for Sustainable Development, DE

89. Ramsha Munir, Technical University of Darmstadt, DE

90. Sandeep Narayanasetti, Indian Institute of Tropical Meteorology, Pune, IN

91. Ousmane Ndiaye, National Agency for Civil Aviation and Meteorology, SN

92. Carlos Nobre, Science Panel for the Amazon, BR

93. Achim Oberg, Professor, University of Hamburg, DE

94. Riko Oki, Director, Earth Observation Research Center, JAXA, Tsukuba, JP

95. Tuba Özkan-Haller, Professor and Dean, Oregon State University, US

96. Tim Palmer, Royal Society Research Professor of Climate Physics, University of Oxford, UK

97. Stan Posey, NVIDIA, US

98. Andreas Prein, National Center for Atmospheric Research (NCAR), US

99. Odessa Primus, Global Arena Research Institute, CZ

100. Mike Pritchard, NVIDIA / University of California - Irvine, US

101. Julie Pullen, Propeller Ventures, US

102. Dian Putrasahan, Max Planck Institute for Meteorology, DE

103. Johannes Quaas, Professor, Leipzig University, DE

104. Krishnan Raghavan, Director Indian Institute of Tropical Meteorology - Pune, IN



105. Venkatachalam Ramaswamy, Director, NOAA/ Geophysical Fluid Dynamics Laboratory - Princeton University, US

106. Markus Rapp, Director, DLR Institute of Atmospheric Physics, DE

107. Florian Rauser, Bundesamt für Strahlenschutz, DE

108. Markus Reichstein, Director, Max Planck Institute for Biogeochemistry, DE

109. Aromar Revi, Director, Indian Institute for Human Settlements, IN

110. Sonakshi Saluja, Reiner Lemoine Institute / Alexander Von Humboldt Foundation, CH

111. Masaki Satoh, Professor, Atmosphere and Ocean Research Institute, The University of Tokyo, JP

112. Vera Schemann, University of Cologne, DE

113. Sebastian Schemm, ETH Zurich, CH

114. Christina Schnadt Poberaj, Executive Director, Center for Climate Systems Modeling C2SM, ETH Zurich, CH

115. Thomas Schulthess, Professor ETH Zurich, and director, CSCS, CH

116. Catherine Senior, Met Office, UK

117. Jagadish Shukla, Professor, George Mason University, US

118. Manmeet Singh, Indian Institute of Tropical Meteorology - Pune, IN

119. Julia Slingo, (Chief Scientist of Met Office, retired), UK

120. Adam Sobel, Professor, Columbia University, US

121. Silvina Alicia Solman, University of Buenos Aires / CONICET, AR

122. Jenna Spitzer, PhD Candidate, Utrecht University, NE

123. Detlef Stammer, Professor, University of Hamburg, WCRP JSC Chair, DE

124. Philip Stier, Professor, University of Oxford, UK

125. Thomas Stocker, Professor, University of Bern (former IPCC WG1-Cochair), CH

126. Sarah Strock, Hewlett Packard Enterprise

127. Hang Su, Institute of Atmospheric Physics - Chinese Academy of Science, CN

128. Petteri Taalas, Secretary General, World Meteorological Organization (WMO)

129. John Taylor, Commonwealth Scientific and Industrial Research Organisation, AU

130. Susann Tegtmeier, Professor, University of Saskatchewan, CA

131. Georg Teutsch, Professor, UFZ - Leipzig, DE

132. Adrian Tompkins, International Centre for Theoretical Physics - Trieste, IT

133. Uwe Ulbrich, Professor, Institute of Meteorology - Freie Universität Berlin, DE

134. Pier Luigi Vidale, Professor, NCAS / University of Reading, UK

135. Chien-Ming Wu, Professor, National Taiwan University, TW





136. Hao Xu, Tencent Sustainable Social Value (SSV) / Tencent Carbon Neutral Lab Tencent, CN

137. Najibullah Zaki, Technical University of Darmstadt, DE

138. Laure Zanna, Professor, New York University, US

139. Tianjun Zhou, Professor, Institute of Atmospheric Physics - Chinese Academy of Sciences, CN

140. Florian Ziemen, German Climate Computing Center (DKRZ), DE

*Author contributions.* All authors: Conceptualization, Writing – review & editing, Bjorn Stevens: Supervision, Writing – original draft preparation, Methodology

*Competing interests.* The authors declare no competing interests.