# Peer review of "Earth Virtualization Engines (EVE)"

_Earth System Science Data, 2023_

## Referee Comment (RC1)

Interesting to see ESSD take on manuscripts describing authors' 'vision' rather than their data. Will other groups seek similar exposure? Those problems remain for future authors and editors. For now, we want this first one to set very good example.

Fundamentally, I applaud authors efforts to push climate community forward. Our community needs and deserves such a push. Without changes such as described here, community will flounder, eventually (if not already) proving irrelevant to most research. I recommend publication in ESSD.

For this reviewer, the manuscript misses key audiences.

First, however, we must deal with AI ('artificial intelligence' in authors' words), which I label instead as 'machine-assisted discovery and learning' (MADL). I adopt generic terminology to make a fundamental point: our data-rich (data-overwhelmed?) science explored these tools and depended on these technologies for a very long time. This reviewer encountered neural networks back during the 1980s. AI emerged recently in social and political spheres but oceanographers, confronting rich but periodic geographically-isolated data, have long applied advanced statistical techniques (random forests, neural networks, automated cluster analyses, etc.). ESSD published, in 2018, a MADL study of ocean carbon states (https://doi.org/10.5194/essd-10-609-2018). More recently, many ESSD publications, often exploring satellite data within Google Earth Engine (GEE), describe MADL applied to land cover changes in challenging regions. Instead of treating AI as new and potentially frightful, EVE authors should suggest positive ways forward. What benchmarks? What inter-comparisons? ESSD (with GMD) attempted a special issue focussed on MADL benchmarks, which brought forward a tropospheric ozone example. What data will prove most in need or most amendable to AI? Should we start from air quality, ocean mixing, satellite NDVI, anthropogenic CO2 emissions? This group - who better - should make recommendations. Otherwise readers gain nothing beyond additional cautious mention of AI. Readers do not need to find exact plans but need assurances of valid approach. Inside, outside, on-top, underlying? Instead of platitudes, readers will want to learn how EVE community intends to help us all adopt and use MADL to increase access to high quality certified data. Fast moving technology, no doubt, but our community has long history of finding, testing and applying these tools.

Many readers will respond positively to EVE mantra of "entirely new and inherently better" and to its laudable social justice intents. But, same readers, finding EVE white paper in ESSD, will wonder about data, education, young researchers, etc.

Technical innovation? Good. But what will EVE provide to researchers at their desks? Easy access? Better documentation? Recognizable formats? One hopes for all of the above, but one finds little information. Will all data reside at one EVE repository, such as used for CMIP? Doubtful, and probably unworkable. If distributed repositories, will EVE finally solve interoperability? 'FAIR' tried but failed; how does EVE propose to approach this challenge? Use GEE approach? If so, how will EVE assure provenance(s)? For researcher who might need historical, present and future ocean surface winds, plankton abundance, and mixed layer carbon export, will they turn to EVE for best reliable information? EVE might aspire to serve as that source but most readers will need assurance that EVE understands the need and proposes valid approach to solve the problem. "Integrate observations" via "digital commons"? Most ESSD readers have read those phrases too many times. Other than good intentions, anything new or different, here?

EVE hopes to "train and employ new developers and users of climate information globally". Good, very positive. Many readers, however, will again have read such laudable intentions before. What does EVE propose to do differently? How will EVE crack publish or perish mentality that haunts (or, discourages) many young researchers? Despite aged luminaries

among co-author list, this reviewer and many readers wonder: why did such an initiative not emerge from young scientists? From YESS, for example? Too many young researchers drop out. Others, having survived initial years, find few reasons for change. What new education, enticement, enrichment or employment models might emerge from EVE? Must we address past educational failures in order to achieve essential equity and climate solutions? This reviewer and many readers suppose 'yes' to that question but we fail to find helpful directions or solutions among EVE documentation.

I envision short example: scientist involved in managing or remediating a mountain stream. Mountain stream might flow from Rocky Mountains of North America, from Alps of Europe or from Tibetan Plateau of Asia. Analysis effort might focus on short upstream stretch but stream will join larger stream will join river might join larger river to eventually deliver sediments, nutrients, contaminants (perhaps including radionuclides) to ocean. Upstream, our researcher might confront atmospheric deposition, acid runoff from mine tailings, agricultural and village waste inputs, relict channelization, fish ladders, etc., and may employ drone-based remote sensing on meter to kilometer scales. That researcher may know ESSD for its descriptions of global streamflow data or for descriptions of and access to relevant data on comparable streams, and perhaps even for riverine terms in carbon or nitrogen cycles. They may calculate seasonal snow-melt inputs from (someone else's) hydrological model forced by downscaled precipitation data from a national GCM. For their own use and for information relevant to their user community, they need (at minimum) local data on present and future (one or two seasons) streamflows, nutrient regime, contaminant sources, benthic productivity and nutrient recycling, bed flows, tree falls, channel engineering, etc., as well as on past protection or logging. Farther upstream, a helpful environmental group might introduce beavers. Colleagues might monitor riparian land forms or local forestry or agriculture practices. Her or his observed catchment might include grassland or forest fires. They will undoubtedly worry about funding; securing and ensuring funding might represent key portion of employment responsibility. If, within the interest zone of ECMWF and ECMWF's climate change services, will our researcher use reanalysis data? In a different region, might they turn to NOAA? What does EVE offer to this (these) researcher(s)? How do they know about EVE? If they glanced at this initial description in ESSD, will they have shown interest? Not without substantial revisions, in view of this reviewer.

Sorry for so many questions. Particularly as applied to an initiative that I basically support, that I think climate community needs. But, for this reviewer and for many readers, EVE resembles only the latest of long line of well-framed well-intentioned but unfunded initiatives. Apparently, EVE considers itself unique among the research community plus, as a consequence, the holder and source of unique solutions. Good! But, provide sufficient detail to prove your point? A more-focused plan, built around tangible example(s), seems much needed.

---

## Referee Comment (RC2)

This article can justly be called a vision, although it already contains some of what could be identified as elements of a proposal, such as concrete deliverables or aspects of funding. Reviewing a "vision" (which may become a full proposal anytime soon) should certainly be different from reviewing a "normal" (ESSD-)article. Due to this, and the obvious merits of the vision (see below), I recommend publication, and leave it to authors whether to incorporate response to critical observations below in the final paper (or perhaps address it when writing their proposal).

The authors are to be commended for presenting a vision which is audacious not only in its financial dimension but as well in the ambition to serve society from the global to the local scale. They strive to provide not only climate projections of "much greater fidelity, with local granularity" (line 19), but the technical basis ("software and infrastructure-as-a-service", line 28) for local communities to derive the impacts of climate change to their living conditions, around the world ("a world where everyone knows how climate and climate change affect them, and where this knowledge empowers them to act." line 14/15 and hint at the scope of projections – from hydrology, agriculture to biodiversity- in lines 55 ff).
The broad diversity – along many dimensions - of "summit participants", line 159, and authors' affiliations support the ability of this group to judge whether it is realistic to deliver on this vision.

Furthermore, it clearly identifies ("Efforts to know more, and plan better, are severely under-resourced, and inequitably distributed" line 11) and strives to address the problems of equitable access to these capabilities. In particular, the vision requires each EVE to "establish and maintain equitable access to a space of interoperable data and software" (line 35) and to "include a strong component of well-tailored capacity development, … to bridge divides to train and employ new developers and users of climate information globally." (line 42/43) and "Ideally EVE would give every country in the world the capability to train people to develop models, AI algorithms, and tailor climate information to meet their needs." (line 100/101).

The visionaries are aware of the broad range of societal needs (lines 55-59) and describe some of the challenges EVEs would face (such as their apparently being in competition with existing organizations, line 135 ff).

However, some of those challenges faced by EVEs are just mentioned or hinted at, leaving the reader guessing (or hoping), how they would be addressed. The three major topics this reviewer sees are:

- As authors explicitly refer to the principles of ESSD (lines 173-176), a lack of clarity about the commitment of EVEs to openness of their outputs must be noted: "open delivery of value", line 51, is a bit too broad, if not vague.
  Is there a clear commitment of EVE's envisioners that all of it outputs - texts, data and software - are to be openly accessible, *in a timely fashion*? Just as crucially, how would scientists from "all countries" be allocated equitable access to the computing infrastructure?

  Leaving these topics to be resolved by "Global governance" under the terms of an "international treaty", line 49, might results in a – from a scientific standpoint -

mindless struggle for each country's piece of the pie and perhaps even to suppression (or delay) of uncomfortable truths.

- The matter of trust (trustworthiness?) or quality/"fidelity" of projections is named, but not addressed by consequences in this article. It would help if some required measures – such as curation of software and data, (organization of) systematic reviews or model-model and data-model intercomparisons - would be named as regular tasks of the EVEs. Methods chosen for systematic scrutiny may need to be unfamiliar, particularly if EVEs would attract a major part of the "projection community" (See the methods of the High Energy Physics community which clusters around the powerful instruments of the CERN LHC.)
The necessity of such measures is evident from the FAQ notes on surprises (lines 64-73) and also when generative AI, well known for hallucinations and going MAD (Model Autophagy Disorder), comes into play (line 104/105).

  Moreover, even those surprises themselves would only be trusted if not just the models are deemed trustworthy, but the observed data stem from similarly trusted parties (Authors seem to note that there is a major gap between something being actually trustworthy and it being finally trusted). Why not suggest that at least some essential data be co-located with EVEs, and treated to the same standard of fidelity?

- EVEs are described as three to five "centers of excellence", and it might appear to readers that a concentration of physical and human resources in just as many locations is meant to be implied. But the discussion about power needs reveals that is not necessarily so ("The compute resources need not be sourced from a dedicated site" line 128). However, federated systems, popular as they are for matching funding structures and not hurting current operators, still have to prove that they are indeed effective and efficient.
(E.g., a recent attempt to port a large-scale analysis from Google Earth Engine to EOSC, paid for by EOSC, failed quite miserably, while the observation that a growing share of ESSD articles is based upon the Google Earth Engine might indicate that scientists indeed flock to an infrastructure which is powerful and convenient at the same time.)

  Also, new co-operation between large numbers of scientists from different scientific domains, different continents and cultures etc. will be hampered by physical distance. Since authors refer to "international organizations whose profile overlap with parts of EVE's remit" (line 132/133): To which extent are those institutions distributed?
Even having just 3-5 EVE sites and to manage that those stay interoperable and do not develop on divergent paths will be challenging (if that is a goal of the vision??).

Some individual observations:

- It appears that EVEs will hold just models (and a copy of some reanalysis data which is used as boundary condition?) The need for access to observed data (remote or in-situ) is acknowledged ("observed changes behave differently" line 71) but the means to achieve that remain vague while hinting at obstacles ("catalyse" line 15, "embed"

line 18, "link" line 20, "strengthen these services by making it easier to access" line 77)
- The answers to "Aren't there already a great number of digital twinning activities, what makes EVE different?", line 89, and "How was EVE's budget estimated?", line 131, omit to mention the elephants in the room, commercial actors such as Google Earth Engine and Microsoft Planetary Computer, particularly Microsoft ClimaX. To defeat a purely economic argument about potentially lower cost, publicly built and operated EVEs need to be at least similarly attractive as those (ease of use, efficiency).

---

## Author Comment (AC1)

**'Earth Virtualization Engines (EVE)' – Author's response to comments**

January 6, 2024

The reviewers are thanked for their thoughtful and constructive comments. Adequately addressing these comments will considerably improve the articulation of EVE. Because the body of the 'Summit Statement' was negotiated over several weeks with the nearly 140 co-authors, we are inclined to leave the main text untouched as a reflection of the Summit, and propose to address the Reviewer Comments through a modification of the FAQs. We note that the FAQs are included as part of the main submission, rather than hidden supplementary material, and no comments were raised about the statement that we felt could not be adequately addressed through the FAQ section.

**Summary of our understanding of the main issues raised by the reviewers**

Here we first briefly summarize what we understand to be the main points raised by the two reviewers, noting the overlap on some issues. In referring to the reviewers specific comments we use the notation ¶$n.m$ to denote the $m$'th paragraph of Reviewer $n \in \{1, 2\}$.

1. More nuance and detail on the topic of AI (¶1.4)

2. Ensuring quality of service and open access to data and software (¶2.8-9)

3. More specificity on capacity development (¶1.5, ¶2.6-7)

4. More specificity on how EVE will effect the scientific culture and opportunities for scientists and scientific development (¶1.5-7, ¶2.6-7)

5. Concretization (¶1.8-9, ¶2.10-11) including meta point 'laudable statements but can we make them tangible' throughout, and how we envision a federation.

Following this itemization, ur approach to addressing the comments is elaborated on below. Before doing so we note that V. Bouchet and D. Stammer have been removed from the list of authors.

**1. More Nuance and detail on AI:**

Among the authors are a considerable number of world leading experts in the area of AI, both from the computational science and geoscience domain. Many of us also share the reviewer's experience of coding our first neural networks in the late 1980s and early 1990s. The change that we, and EVE, respond to is an appreciation that the new capabilities of AI, particularly generative AI, come from the size of the data sets, the size of the AI model, and the availability of computational power that enables to set the parameters of the latter given the former. This is the phase change. EVE responds by proposing to create massive amounts of data (projections)

at scale that can be linked to auxiliary (external) data to train massive models through compute resources co-proximate to the main data streams. In this respect, EVE nodes would constitute a federated landscape of what some are calling AI Research Resources.

*We would address this more directly through changes to the FAQs 2, 3 and especially 6.*

**2. EVE's effect on the scientific culture:**

EVE would influence the scientific culture and practice in three ways. On the side of data provision, by greatly increasing the use of climate data (including model projections) it helps define scientific priorities that would improve data quality, either by improving the data collection, or its generation through improved modelling (also on regional scales). On the other side, by defining protocols that link the EVE nodes to lighter, distributed, data sets, enabling users to link this to their own (perhaps proprietary data), and by creating community access to the co-proximate computing and data for data that is not distributable (due to is shear amount, i.e. km-scale projections, and some satellite data) it would create a community resource that gives scientists, and climate service providers access to data and resources that they would not otherwise have. Finally, by constituting this as a professional activity that stimulates research, EVE creates opportunities that help circumvent some of the obstacles (e.g., academic metrics of personal achievement) noted in ¶1.7.

*FAQs 7 and 11 would be rephrased and their answers revised to make these points more directly.*

**3. More specificity on capacity development:**

EVE's aspiration to strengthen capacity development, etc., is mentioned in several places (i.e., FAQs 4 and 5), but the reviewer's comments highlighted how the presentation lacked specificity. Involving communities in the information creation EVE would facilitate is not just a noble aspiration, it is materially necessary. The need for capacity development has two aspects (i) EVE needs access to local data, and only by engaging those responsible for these data can it create the social process needed to valorizes local data collection and provision; and (ii) to ensure that the information that comes from EVE is trusted and actually useful and used by the communities it targets, the communities that are expected to use the data must be involved in its creation (presently in FAQ 5). This makes the capacity development organic to EVE, and further justifies the idea of EVE as a federation of *regional* center's of excellence.

*We propose reformulating and reframing FAQ 5 which was meant to address these aspects, but did so indirectly and was ineffective. The reformulation would incorporate feedback from the presentation of EVE at the WCRP Open Science Conference in Kigali, that specifically addresses how EVE can (and must) strengthen capacity development for it to be successful. We would also modify (or delete) FAQ 12 given that in the meantime both COP and the OSC (Kigali) meeting have transpired.*

**4. Quality control of data and software:**

EVE would address issues related to quality assurance, trust worthyness, etc, in three important (and largely novel) ways. Foremost for data quality/trustworthyness is that: (i) the data is used; (ii) that its creation is transparent; and, (iii) that its creation and provision is responsive to those that use it. EVE's novelty becomes apparent when one considers how the second and third points are addressed in our existing approaches. Meeting these goals requires an institutional framework to sustain a feedback loop that makes the data provision responsive to its use. We propose to add a bullet on this point, which also mentions the role of AI (i.e., if generative AI is used to convert data to information how do we guard against hallucination, etc.).

*We propose a new FAQ, to address this point directly, making the above points and introducing the importance of EVE's Centers of Excellence to the concept as a whole.*

*5. Concretization:*

The issue of being more specific resonates throughout the reviewer's comments. We originally tried to address this by the FAQ on what would happen without EVE. In the revisions we sharpen the presentation of how EVE differs in concrete ways from existing approaches, emphasizing the importance of regional Centers of Excellence taking the form of actual institutions federated internationally to ensure accountability, capacity development, professional opportunity, and societal acceptance.

*We propose to rework the (now) final FAQ to address these issues, and update the next steps as a final FAQ to address the importance of an international framing for many of the above stated objectives (e.g., capacity development, standard development, information acceptance and distribution.*

---

## Author Response (AR1)

**'Earth Virtualization Engines (EVE)' – response to reviews**

**February 11, 2024**

A synthetic response to the reviews was provided in the "author comments" uploaded on 6 January 2024. Here we supplement this with a brief response outlining the changes to the manuscript in response to the reviewer comments.

**Review 1**

The substantial points raised in this review were presented in a sequence of paragraphs, starting with paragraph 4. These are addressed in turn below.

1. "First, however, we must deal with AI ..." The FAQs have been substantially revised to address this point, as AI is central to EVE. EVE differs from what has been done in the past and what is being done elsewhere by its scale, and this, we have learned, is decisive. Specifically FAQ2 (they are now numbered) references the realization that generative AI benefits from the scale of the problem "which provide the vast amounts of globally gridded data needed to train the most advanced weather- and climate-related AI models ... [EVE] will demonstrate how interoperability of data and software co-proximate to the computational capacity needed for the requisite machine learning, can help users extract more information, with greater salience, from their own data, thus valorizing its collection and open provision". In FAQ4 we add "Third, whereas the aforementioned efforts mostly focus on twinning as a form of data provision, EVE additionally emphasizes AI integration to enable information provision." FAQ6 has been substantially revised to address the reviewers point. It now concludes: "AI On-Top goes well beyond emulation to enable interactivity across disparate sources of data, e.g., to create new types of models, and to give salience to the use of data. This is the game changer. One lesson of recent applications of generative AI is the disproportionate benefit of larger (AI) models, trained on larger and more diverse data. This requires very large computing capacity, both for the training and for creating the training data. It means that AI, in particular AI On-Top, isnt merely a part of EVE, it is one of the main reasons why EVE is necessary."

2. The first bullet of the summit statement itself has been modified to emphasize the importance of quality control. This is also emphasized through the addition of FAQ12 'How would EVE ensure data quality and accessibility'. Education and young researchers are included in the revision to the 'what next' (FAQ14) as well as to FAQ11 which addresses the trap of publish or perish and career development, and through the example of CERN in FAQ5

3. "But what will EVE provide to researchers at their desks?". We have addressed this by considerably expanding FAQ7, also outlining new uses of regional models. It was and is further discussed in FAQ13, on what would happen without EVE, where the advantages of standardizing bodies is raised. Already in the original we mentioned how EVE would give

researchers around the world greater access to computing capacity, and how EVE would create a demand for scientists, i.e., job opportunities (FAQ11). These changes also address the next paragraph ('EVE hopes to . . . ') of comments in Review 1.

More generally we agree with the Reviewer's comments about the need for specificity. While we have made some efforts in this regard in the revisions of the FAQs, particularly to better emphasize the interactivity. We acknowledge that these are rather superficial changes to the substantial point. In our defense we believe that the substance of this point is better developed outside of this statement. Toward this end we have relaunched the EVE website , and are in the process of developing much more specific and concrete proposals that align with existing activities and funding opportunities. This is however not something we are ready to address collectively and in this first statement, especially give the constraints of a document with nearly 150 names.

**Review 2**

The points raised in the review were itemized in a bullet list, which we address in turn.

1. On openness and capacity development. Yes, the point of EVE is to expand the public space in the digital sphere, which by definition is an aspiration of openness. The FAQs have been modified to say this more specifically. We now mention the aim of capacity development in FAQ1, and the reference to CERN's role in developing European postwar capacity in FAQ2, as well as the essential need for EVE to engage local stakeholders. A new FAQ (12) has been added to address data quality and global governance is identified as only one of the governance options. Specifically we say "This could be in the form of an international treaty, or through coordination of self-governed centers by existing intergovernmental organizations." This has been slightly revised to avoid anticipating specific institutions as playing a role in coordinating EVE.

2. The matter of trustworthiness is addressed in the aforementioned and new FAQ12. Hallucinations are specifically addressed by this revised FAQ. Also the importance of users being involved in the creation of the information is recognized as an important basis for developing trus and is raised through modifications to FAQ5.

3. We leave this issue of a sharper presentation of EVE's structure a bit open, as part of the forward looking revision to 'What's next' (FAQ14). Our expectation is that, if we are successful, the way in which regional centers of excellence will be built will differ regionally. The comments raised by the review are nonetheless prescient. We don't have answers to them all, but we also don't think that they need to be answered yet, by us, as long as the principle of interoperability and coordination is maintained.

4. We try to be more specific that EVE will also host data. Here however, as seen in the changes to FAQ2 which addresses this point, we are sensitive to the fact that many groups want to maintain control of their own data, and for the case where this data is light weight, it can be linked to the EVE centers with the help of standardization and protocols. EVE's bet is that by valorizing data, it will be more widely collected and shared.